# OUMG: Objective and Universal Metric for Text Generation with Guiding Ability

## Abstract

Existing evaluation metrics for text generation rely on comparing candidate sentences to reference sentences. Some text generation tasks, such as story generation and poetry generation, have no fixed optimal answer and cannot match a corresponding reference for each sentence. Therefore, there is a lack of an objective and universal evaluation metric. To this end, we propose OUMG, a general metric that does not depend on reference standards. We train a discriminator to distinguish between human-generated and machine-generated text, which is used to score the sentences generated by the model. These scores reflect how similar the sentences are to human-generated texts. The capability of the discriminator can be measured by its accuracy, so it avoids the subjectivity of human judgments. Furthermore, the trained discriminator can also guide the text generation process to improve model performance. Experiments on poetry generation demonstrate that OUMG can objectively evaluate text generation models without reference standards. After combining the discriminator with the generation model, the original model can produce significantly higher quality results.

## 1 Introduction

According to different expected goals, natural language generation tasks can be divided into two categories. The first type of task has a specific reference standard – usually given by experts. The goal of model optimization is to make the generated results close to or even coincide with the reference standard, such as machine translation (Koehn, 2009) (Bahdanau et al., 2014) (Vaswani et al., 2017), summarization (Chopra et al., 2016) (Rush et al., 2015) (See et al., 2017), and image captioning (Fang et al., 2015). The existing metrics have been able to evaluate these tasks from various aspects and obtain objective scores. The second type of task includes dialogue system(Kurach et al., 2016), story generation (Fan et al., 2018) (See et al., 2019), poetry generation (Lau et al., 2018) (Liao et al., 2019), etc. The ultimate goal is to make the model learn the features of natural language and generate sentences that look like written by humans. These tasks have a common characteristic: The suitable reference answer is not unique (e.g., The same beginning can be followed by various story plots). Therefore, the evaluation will inevitably be biased if only a few sentences are used as a reference standard.

In order to better evaluate the quality of text generation results, a universal evaluation metric must meet the following three requirements: 1. The metric must be available in the absence of corresponding reference standards. 2. In the same semantic, the change of expression and words should not significantly affect the evaluation score. 3. The metric must be objective.

In this paper, we propose OUMG, a more general evaluation metric for text generation. This method meets the above three requirements and has the ability to guide the generation process, which can further improve the quality of sentences generated by the model.

The core of our approach is a discriminator, whose function is to distinguish between human-generated and machine-generated text. Ideally, this discriminator should be trained with a set of sentences written by humans as positive samples and sentences generated by various models as negative samples. In order to ensure universality, the selection of negative samples must cover all the potential models. However, under realistic conditions, this is impossible. Therefore, OUMG uses a compromise by mixing the generated results of the two models(e.g., The improved model and its baseline) equally as a negative example. After sufficient training, the new samples generated by the

two models are scored by the discriminator, and the relative merits of the two models are judged according to the high score rate difference and upgrade rate. In addition, a discriminator with high accuracy can consider the semantic information of the sentence, avoiding the problem of large fluctuations in scores caused by different expressions. At the same time, a model trained by datasets is more objective than human judgments, which meets the requirements of an evaluation metric.

For the sake of showing the unique advantages of OUMG, we choose poetry generation as the main task in our experiments. But this does not mean that OUMG can only be used for the second type of task. Since we eliminate the dependence of metrics on reference standards, our method is universal and can also be used for the first type of task. For example, in machine translation, we can use the original sentence and its corresponding translation as the input of the discriminator and train the discriminator to distinguish whether the translation part of the input comes from the reference standard or the machine translation model. Other tasks are similar.

Inspired by Yang & Klein (2021), our method can also be used to guide text generation. In the text generation process, we use the discriminator to score all the candidate tokens for each step and combine the score with the logits provided by the generator to select the tokens with higher final scores. Compared with the Yang & Klein (2021) method, we add a hyper-parameter $\alpha$ to control the influence of the discriminator on the generation process.

## 2 RELATED WORK

### 2.1 METRICS BASED ON STRING MATCHING

The earliest metrics use the degree of string matching between two sentences to measure whether the candidate sentence is close to the reference standard. The representatives are BLEU (Papineni et al., 2002) and ROUGE (Lin, 2004) , which were initially applied to machine translation and were widely used in various other tasks in later studies. The idea is to count the number of times the same word appears between two sentences. If two sentences have a higher degree of coincidence, they are more similar. The difference is that BLEU mainly focuses on the accuracy of the text generation model, while ROUGE focuses more on the recall. There is an obvious problem in this method: only the most superficial matching can be carried out, and no semantic information can be taken into account. It means that these metrics will equate synonyms with completely irrelevant words and are not sensitive to the word order of sentences, which severely limits their accuracy.

### 2.2 METRICS BASED ON WORD EMBEDDING

In order to solve the problem that the metrics based on string matching cannot consider semantic information, researchers began to propose metrics based on word embedding. The original method is to convert the sentence into vector representation by the Word2vec models (Mikolov et al., 2013) and then calculate the similarity between the two sentences by the cosine similarity method. After word embedding, the generated vector has expressed and abstracted the semantic to a certain extent so that different words with the same meaning have a closer distance. Thereforesuch metrics scoring has more reference value. With the development of many pre-training language models represented by Bert (Devlin et al., 2019) , more people use Bert and other models (Peters et al., 2018) to replace the traditional Word2vec models. In contrast, Bert and other models can more dynamically capture the context information, solve the polysemy problem, and refine the semantics more accurately. Bertscore (Zhang et al., 2020) and BLEURT (Sellam et al., 2020) are representatives of such metrics. However, although the metrics based on word embedding can well combine semantic information, they still rely on authoritative reference standards, which are only applicable to the first type of text generation task.

### 2.3 METRICS BASED ON LANGUAGE MODEL

The above two metrics evaluate the performance of the model through the sentences generated by the model. In addition, the language model itself can also be assessed directly. The PPL metrics evaluates whether the language model can generate smooth sentences by calculating the joint probability. Given a trained language model, use it to calculate the probability of each sentence in the test set. The higher the likelihood of this sentence, the lower the corresponding PPL score. That

is, the better the modeling ability of the language model. Although many text generation tasks use the PPL score as a reference for model quality, there are still some unavoidable problems. First, the PPL metric only evaluates the probability of the output of the language model without considering the impact of the optimized decoding process on the final production, so it can only be used to evaluate the modeling ability of the language model. In addition, PPL only focuses on the fluency of the sentence itself but ignores other factors that affect the quality of the generated text, such as the consistency with the context and the correspondence with the given topic. It cannot effectively evaluate the quality of the text in some tasks, like dialogue generation and poetry generation.

## 3 OUMG

### 3.1 DISCRIMINATOR

The function of the discriminator is to identify which are written by humans and which are generated by the models in a set of sentences. The discriminator outputs scores from 0 to 1 for each sentence, which is the probability that the sentence is judged to be human-generated. When the score of a sentence is higher than 0.5, we classify the sentence as human-generated.

The following formula can express the loss function of the discriminator:

$$l(\theta) = -\sum_{x \in X_1} log(D_\theta(x)) - \sum_{x \in X_2} log(1 - D_\theta(x)) \tag{1}$$

Where $N$ represents the discriminator with $\theta$ as the parameter, and represent the human-generated sample data set and the machine-generated sample data set, respectively.

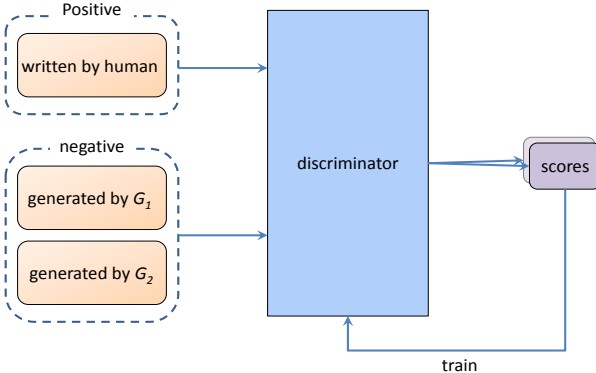

Figure 1: Training process of the discriminator

The trained discriminator will be used to score a sentence. Since some text generation tasks lack standard reference, we use relative metrics to avoid this problem. The OUMG metric is used to compare the sentences generated by the two models and measure the gap between them. In practical application, we need to compare the two text generation models $G_1$ and $G_2$, and obtain the relative scores of the two models. To ensure the fairness of the two models, when generating the sample data set, we stipulate that sentences generated by the two models are mixed equally, thus (1) becomes

$$l(\theta) = -\sum_{x \in X_1} log(D_\theta(x)) - \sum_{x \in X_{G_1}} log(1 - D_\theta(x)) \sum_{x \in X_{G_2}} log(1 - D_\theta(x)) \tag{2}$$

After fully trained to converge, the discriminator can distinguish human-generated and machine-generated sentences with high accuracy. The higher the sentence scores, the closer it is to human-generated sentences. Next, we use the trained discriminator to score two test sets from the model $G_1$ and $G_2$ and get two groups of scores. We process the score of the discriminator adopting the following two relative metrics:

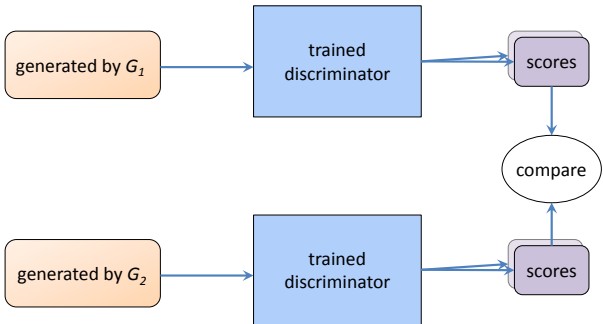

Figure 2: Process of evaluation using the discriminator

**High Score Rate Difference(HSRD):** When a sentence gets a score of more than 0.5 after it is input into the discriminator, we call it a high score, indicating that the discriminator determines that the sentence is written by humans. High score rate(HSR) refers to the ratio of a set of sentences that are judged as high score to all sentences in a model and high score rate difference is expressed as

$$\frac{N_{D_\theta(x_{G_1})>0.5} - N_{D_\theta(x_{G_2})>0.5}}{N} \tag{3}$$

Where $N$ represents the total number of sentences generated by each model, $N_{D_\theta(x_{G_1})>0.5}$ and $N_{D_\theta(x_{G_2})>0.5}$ represent the number of high-score sentences in the two groups. To avoid the interference of redundant factors, the same number of sentences generated by each model is required in the comparison.

For the two-generation models $G_1$ and $G_2$ to be compared, if the high score rate difference is a positive number $s$, the probability that model $G_1$ generates realistic-looking text is $s$ more than that of $G_2$ . If $s$ is a negative number, the conclusion is the opposite.

**Upgrade Rate(UR):** Given the same set of inputs, two models, $G_1$ and $G_2$, to be compared are used to generate the sentences, and the discriminator is used to judge the generated sentences. By comparing the scores of the output under the same inputs, we get the upgrade rate. The specific formula is shown in (4):

The specific formula is shown in formula (4)

$$\frac{N_{D_\theta(x_{G_1})>D_\theta(x_{G_2})}}{N} \tag{4}$$

Where $N_{D_\theta(x_{G_1})>D_\theta(x_{G_2})}$ represents the number of times $G_1$ scores more than $G_2$. Similar to the high score difference, it is also stipulated here that the number of sentences generated by each model is the same, namely $N$.

The upgrade rate indicates the probability that $G_1$ generates better sentences than $G_2$ when the input is the same. When the upgrade rate is a positive number close to 1, we can assume that $G_1$ outperforms $G_2$ in most cases.

### 3.2 GUIDE TEXT GENERATION

Another critical feature of OUMG is to adjust and guide the text generation process to make the generated results closer to the sentences written by humans. Our inspiration comes from FUDGE Yang & Klein (2021) and has been improved on this basis. The process that guides text generation is shown in Figure 3:

Given an autoregressive language model $G$, the model can output the probability $P(x_i|x_{1:i-1})$ of the next word under the condition of a given word sequence. Limited by the performance of the

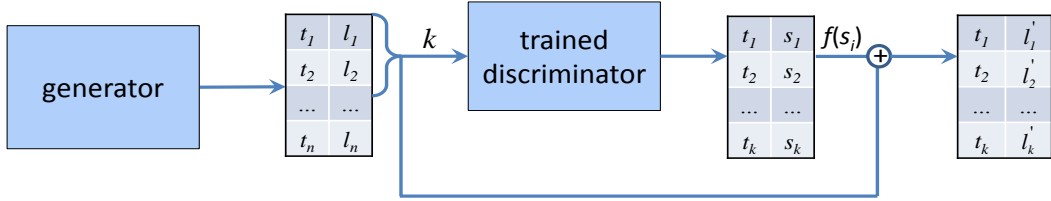

Figure 3: Guiding text generation with discriminator

text generation model, the probability distribution of the model is not the same as that of the actual natural language, which leads to the bias of the above conditional probability. As the generated text gradually grows longer, the bias will become larger and larger. OUMG changes the probability distribution of the final output by scoring the candidate with a discriminator at each step, making it closer to the probability distribution of natural language, thus alleviating the bias.

When guiding text generation, the discriminator we use has the same structure as the discriminator for metric. The only difference is that the data set used for the training discriminator is different. Since the autoregressive language model is generated word by word in order, the discriminator for guiding generation must have the ability to score an incomplete sentence. We change the training set based on the discriminator for metric, split each complete sentence into fragments of different lengths, and put the same label on it. After sufficient training, the generator can judge the similarity between the final generated sentences and human language in advance when the sentence is not completed and give a probability. Next, we will use this probability to guide the following generation process.

Assuming that there is no bias, the output $P(x_i|x_{1:i-1})$ of the generator is the actual probability. When a bias occurs, a lower score is obtained after the candidate sentences are input into the discriminator, and the greater the bias, the lower the score. Therefore, we use the discriminator to adjust the logits of the generator. Considering the running time, We select $k$ tokens with the highest probability, and combine each token separately with the currently generated sentence fragment $x_{1:i-1}$ to input into the discriminator. To facilitate the calculation, we also use the output logits of the discriminator instead of the final scores, which is represented by $s_i(1 \leq i \leq k)$. Next, $s_i$ is processed as follows :

$$f(s_i) = \alpha \frac{(s_i - s_{min})}{(s_{max} - s_{min})} \tag{5}$$

Where $\alpha$ is a hyperparameter, the purpose of which is to facilitate more flexible control of the impact of the discriminator score on the final result. We add $f(s_i)$ to the logits of the generator to obtain new logits for the following sampling.

## 4 EXPERIMENTS

### 4.1 EVALUATION METRICS

The main experiments of this paper is to generate Chinese poetry. Chinese poetry has different rules from modern language, and many people do not understand them well. In addition to some experts, most ordinary people cannot distinguish between ancient and machine-generated poetry, nor can they objectively evaluate them. Therefore, human evaluation cannot be used as the most authoritative metric. Moreover, generating poetry requires that the generated sentences are diverse and even artistic, so there is no fixed reference standard. Due to the above characteristics, common metrics cannot be used for Chinese poetry tasks, but our method is not affected, which fully reflects the versatility of OUMG.

Discriminators in OUMG can be flexibly selected, with the only requirement being to ensure classification accuracy. Our experiments use an attention-based Bi-LSTM model (Zhou et al., 2016) as the discriminator, which has a relatively simple structure and good performance.

| Model | Acc | HSR | HSRD | UR |
|-------|-----|-----|------|-----|
| $G_1$ | | 0.083 | | |
| | 0.906 | | 0.050 | 0.869 |
| $G_2$ | | 0.033 | | |

Table 1: OUMG is used to evaluate the results of the SongNet model before and after modification. We take the model before the change as $G_1$ and after the change as $G_2$. Acc represents the classification accuracy of the trained discriminator on the test set.

We follow the work of SongNet(Li et al., 2020) and use their generator model, which has achieved better performance than GPT-2(Radford et al., 2019) in the poetry generation task. To prove the effectiveness of the metrics, we need to use two models with apparent quality differences to compare. We take the original structure of SongNet as a better model and modify the decoding part as a poorer model. SongNet uses the top-k sampling method in decoding, which intercepts $k$ tokens with the highest probabilities and then samples them according to likelihood. When each token except punctuation is generated, we shift the sampling range by $4k$ elements, that is, the tokens with the sequence number from $4k$ to $5k$ after sorting by probabilities. Since the original model has successfully modeled the probability distribution of Chinese poetry, when $4k$ words with the highest probability are discarded in each decoding, the quality of the final sentences will inevitably decrease significantly. Therefore, we take the modified model as a poor model to verify whether OUMG is effective.

We use the same dataset as SongNet and take its training set, about 20 000 Song Ci, as positive samples of the discriminator. Next, we let the two models to be compared generate the same number of Song Ci, the two mixed as a negative samples. Using the above dataset to train the discriminator fully, the final test results are shown in Table 1

It can be seen from Table 1 that the correct rate of the discriminator exceeds 90 percent, indicating that the discriminator can distinguish the poetry between generated by two models and written by humans according to their different characteristics. Their high score rate difference and upgrade rate are positive, indicating that $G_1$ outperforms $G_2$. As can be seen from the upgrade rate, model $G_1$ has nearly 87 percent of output sentences better than $G_2$ for the same input. In addition, the high score rates of the two models are minimal, indicating that there are apparent differences between the sentences generated by the two models and the sentences written by humans.

### 4.2 GUIDING TEXT GENERATION

The discriminator used to guide text generation have the same structure as discriminators used as metric but have different training sets. The new training set is generated by truncating the complete sentence word by word to form the set of the first $i$ words of the sentence ($2 \leq i \leq n, n$ is the length of the sentence). We use the processed sentences in SongNet's training set as positive samples and the same number of processed sentences directly generated by SongNet as negative samples. Finally, we combine the trained discriminator with the decoding part of the generator.

Compared with the results in Table 1, the discriminator accuracy in Table 2 is reduced to 85 percent. This is because the sentences used as negative samples are more similar to positive samples, which increases the learning difficulty of the discriminator. The high score rate of $G_1$ is significantly higher than that of $G_2$, and the high score rate difference indicates that $G_1$ is 19 percent more likely to produce realistic-looking poetry than $G_2$.

Compared with Table 1 and Table 2, we find that the same model ( $G_2$ in Table 1 and $G_1$ in Table 2 ) has different high score rate in different evaluation results due to the characteristics of relative metrics. In OUMG, the evaluation score of a model is affected by the comparison model. The score reflects the relative gap between the two models rather than absolute performance evaluation.

| Model | Acc | HSR | HSRD | UR |
|-------|-----|-----|------|-----|
| $G_1$ | | 0.231 | | |
| | 0.851 | | 0.190 | 0.811 |
| $G_2$ | | 0.041 | | |

Table 2: OUMG is used to evaluate the results of the SongNet model with and without discriminator. We regard the model with discriminator as $G_1$ and the model without discriminator as $G_2$.

## 5 CONCLUSION

We propose an objective and universal automatic evaluation metric for text generation named OUMG. This method uses a discriminator to score the sentences generated by the two models to measure the relative gap between the two models. Since the discriminator can distinguish human-generated sentences from machine-generated sentences with high accuracy, the score of a sentence can be regarded as the similarity between the sentence and sentences written by humans, which ensures the effectiveness of the metric. Our method can be more commonly used for text generation tasks without reference standards than the traditional text generation metric. Compared with human judgments, our practice does not prefer specific characteristics and is more objective. The experiments show that OUMG can make a reasonable evaluation of the text generation task without reference standards. Moreover, the discriminator can further improve the performance of the original model by guiding the generation process.

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
