# OpenReview forum: "OUMG: Objective and Universal Metric for Text Generation with Guiding Ability"
_ICLR.cc/2022/Conference — ICLR 2022 Submitted_

### Official Review · Reviewer_27Bu · 2021-10-22

**Correctness:** 1
**Technical Novelty And Significance:** 1
**Empirical Novelty And Significance:** 1
**Recommendation:** 1
**Confidence:** 4

**Main Review:**

The paper has the following weaknesses:
- Given that the goal of the paper is to present an evaluation metric, I would expect a study comparing its output with human evaluation, as done for example in the BertScore paper cited. As it is, all we know is that it can tell between two systems one of which is made to be worse than the other. That doesn't tell us much about whether it correlates with human judgements.
- It is true that human judgements can be subjective, but if this is a concern, then one should consider a task where objective evaluation is possible. However, this is not the case with Chinese poetry generation, or any kind of literature generation in general.
- The equations 1 and 2 have symbols that are not explained, i.e. X_1 and X_2.  Also, in equation 2 I think a plus sign is missing between the last two terms of the loss function.
- I don't think that most natural language generation tasks that have a singe correct answer; even in machine translation there are multiple references given.
- Apart from Yang and Klein,  the use of trained discriminator for guiding natural language generation has been previously explored, e.g. : http://proceedings.mlr.press/v119/scialom20a.html . As it is considered by the authors a contribution of this paper to be able to guide the generation, a comparison is needed against these methods is needed.

**Summary Of The Paper:**

This paper proposes a reference-less evaluation metric for natural language generation tasks. It presents experiments on Chinese poetry generation.

**Summary Of The Review:**

The evaluation is flawed as the proposed natural language evaluation metric is not compared to human judgements. Also, Chinese poetry generation is not a suitable task to evaluate on given its subjectivity.

---

> ### Author Response · Authors · 2021-11-21
> **Response to Reviewer 27Bu**
>
> Thank you very much for your comments. we are sorry to have some writing omissions in the paper, which will be fixed in the next version. In this paper, we try to propose a universal and objective text generation metric.  The reason why we choose the Chinese poetry generation task is that the existing text generation metric is not ideal for the task. We want to prove the generality of our method by that. In order to further improve the persuasion, we will add some experiments on different tasks in the next version of the paper. Thanks again.

---

### Official Review · Reviewer_BR3n · 2021-10-30

**Correctness:** 1
**Technical Novelty And Significance:** 1
**Empirical Novelty And Significance:** 1
**Recommendation:** 1
**Confidence:** 4

**Main Review:**

Strengths: with respect I'm not sure there are any.
Weaknesses:
- The paper is poorly written, with many sections difficult to follow. The proposed metric is named OUMG, which I'd guess is meant to stand for Objective and Universal Metric for Generation, although I don't think it's defined anywhere. Assuming this, both parts (objective, universal) are not accurate. A few examples of the difficulty I had with the writing: section 3.1 eon 1 doesn't defined X1 or X2 (which do appear in equation), but does define N (which doesn't). The upgrade-rate and high-score-rate-difference seem overly complex, for what is essentially one system scores higher than the other. But this is getting into minor details; there are larger flaws with the paper.
- The primary flaw is that it makes grand claims about being a metric suitable for use in many places, yet only reports results on a single poetry task, which further are very weak since they don't involve any ground truths e.g. via human ratings. Indeed the paper even makes the claim that for this poetry that "Therefore, human evaluation cannot be used as the most authoritative metric". This just seems plain wrong. It might need the right humans. But the main point is that there is nothing objective in the results here and the reader can't trust the conclusions as a result.
- Statements like "which fully reflects the versatility of OUMG" are very non-scientific given the contrast with what results are actually reported.
- The "only requirement" of the discriminator is accuracy, yet the paper then uses a bi-lstm, yet all recent work in NLP would suggest you'd build a more accurate classifier with a transformer model. This is slightly nit-picking but there is a constant disconnect between the statements made and the actual method and results. Another example, the bi-lstm is described as having a "relatively simple structure". This simply isn't scientific language and shouldn't be in a paper.
- The discussions at the end of section 3 are both not very clear and seem to muddle what a language model does (score more likely language higher) compared to what's required with poetry (where surprise is required).

**Summary Of The Paper:**

An NLG metric for problems where there is many plausible options for the generated text (e.g. story telling, poetry generation, dialog continuation) is proposed.
Results are reported on one one dataset, but the paper is very poorly written and makes a lot of claims which are not correct.

**Summary Of The Review:**

I read the second half of this paper quickly since IMHO there are major flaws with the work. The paper makes many claims about how widely applicable the proposed metric is, and lacks evidence for nearly all of them.

---

> ### Author Response · Authors · 2021-11-21
> **Response to Reviewer BR3n**
>
> Thank you very much for your comments. we are sorry to have some writing omissions in the paper, which will be fixed in the next version. In this paper, we try to propose a universal and objective text generation metric.  The reason why we choose the Chinese poetry generation task is that the existing text generation metric is not ideal for the task. We want to prove the generality of our method by that. In order to further improve the persuasion, we will add some experiments on different tasks in the next version of the paper. Thanks again.

---

### Official Review · Reviewer_pGB6 · 2021-11-02

**Correctness:** 1
**Technical Novelty And Significance:** 1
**Empirical Novelty And Significance:** 1
**Recommendation:** 3
**Confidence:** 5

**Main Review:**

First, I appreciate the efforts of the authors in devising a simple method towards addressing the evaluation of automated text generation systems. Simpler methods are easier to interpret and reproduce. However, it is also important to properly support scientific claims and methodologies with sufficient experiments. By this, I notice that there seems to be an overgeneralization in the problem statement. The authors indicate that for systems like story generation and poetry generation, the final suitable reference answer is “not unique”. While this is true, it is not the only reason why these tasks are special. There should be further clarification that the freedom of the reference answer is not entirely free in open-ended text generation tasks since there are still some rules involving rhyming, format, and integrity that need to be followed specifically in the case of poetry generation as highlighted in Li et al (2020). The addition of this discussion should be in order since this is the main (and only) task tackled in the paper.

There is no discussion of how the proposed metric discriminates between human and model-generated texts. What textual factors do you look for? Do you check the factual consistency? The readability? The cohesion? The stylistic properties? Likewise, there seems to be no discussion on the two models based on their similarities/differences by nature. Discussion for these parts should be in order, not to mention this is the main contribution of the paper.

There is no definitive review of recent works specifically for the development of objective evaluation for text generation. The only qualified citations are for BLUE and ROUGE, the rest are mostly from types of methods (ex. BERT, Word Embeddings) but not the actual work done for evaluating text generation systems. For starters, I recommend the authors to look at papers of BERTScore (Zhang et al, 2020), BLEURT (Sellam et al, 2020), and WMD (Kusner et al, 2015) for guidance of proper comparison of evaluation metrics across text generation tasks. In addition, I also expected the paper of Lin et al (2020) to be included here but was only mentioned later in other sections.

Technicality-wise, it’s extremely hard to believe some of the claims of the paper (“our method is universal and can also be used for the first type of task”) without empirical results from experiments for both Type 1 and Type 2 tasks and without the use of other datasets to prove sufficient “universality” of the evaluation metric. Even with a high accuracy reported in the paper, the proposed metric has only ever been tested for one task (poetry generation) and one dataset (SongNet). Just to mention, the page limit is 8. The authors are advised to consume all the available space to supply needed experiments and comparisons (including baselines) with text generation tasks.

Lastly, there are numerous instances where parts of the discussion involving experimentation details are extremely vague such as:
1. In Section 3.2, “we trained it sufficiently” --> to what extent? Lacks specific details of retraining.
2. From the same section, “split each complete sentence into fragments of different lengths” --> same question, to what extent? The answer to this is found in the next section but it should already be mentioned in the first instance.
3. In Section 4.1, “Chinese poetry has different rules from modern language, and many people do not understand them well.” --> This is another overgeneralization. What are these “different rules” and why do “people don’t understand them well”? Further clarification is needed.
4. In Section 4.2, the only motivation for choosing Bi-LSTM is “which has a relatively simple structure and good performance” and nothing more. I would like to see a stronger basis for this choice. Provide justification as to why Bi-LSTM is the most practical model for the discriminator for Chinese poetry generation with respect to the task at hand. Cite legitimate references along the way.
5. There seems to be an overuse of the word “significantly” scattered in the paper without proper statistical testing. I suggest revising this to “substantially” as the word is already overloaded in literature.


**Summary Of The Paper:**

The paper describes OUMG, a model-based metric for evaluating text generator systems which does not rely on reference standards or gold-standard data. The model is trained similar to a conventional GAN or Turing test-like setup: a discriminator is trained with human-generated and artificially generated sentences to produce a soft-output (probability) to judge if an input text is human-generated or not. After which, the model can be applied as a guide for generating text autoregressively in the hopes of producing texts of better quality. The metric is tested only on a Chinese poetry generation task with one dataset and a single model architecture (Bi-LSTM) both coming from the SongNet study (Li et al, 2020).


**Summary Of The Review:**

Overall, the current state of the paper needs to be improved to match the authors’ claim on the universal/objective applicability of the proposed metric. I see no strengths from the paper empirically although I particularly favor the simplicity of the proposed method. The major weakness of the paper is the lack of sufficient documentation of the practicality of the proposed metric via rigorous and extensive experiments involving various text generation tasks and more datasets. Thus, I am rejecting the paper based on these grounds and will wait for clarifications from the authors.

---

> ### Author Response · Authors · 2021-11-21
> **Response to Reviewer pGB6**
>
> Thank you very much for your comments. we are sorry to have some writing omissions in the paper, which will be fixed in the next version. In this paper, we try to propose a universal and objective text generation metric.  The reason why we choose the Chinese poetry generation task is that the existing text generation metric is not ideal for the task. We want to prove the generality of our method by that. In order to further improve the persuasion, we will add some experiments on different tasks in the next version of the paper. Thanks again.

---

### Official Review · Reviewer_Nb33 · 2021-11-02

**Correctness:** 2
**Technical Novelty And Significance:** 2
**Empirical Novelty And Significance:** 2
**Recommendation:** 3
**Confidence:** 4

**Main Review:**

The main strength of this paper is suggesting an interesting idea to evaluate generated texts. I strongly agree that most evaluation methods rely on comparing with ground truth data - human-written texts. But unlike classification tasks, the output of natural language generation (NLG) models can be various and appropriate since the language can have different expressions even if they have the same semantics. The idea is quite sound and straightforward so that the other researchers can research the claim and idea.

What concerns me most is the lack of experiments to support the claims in this paper. First, U in OUMG stands for universal, but the authors only show the performance on poetry generation task. As the authors mentioned, we can apply the same idea to other NLG tasks, but we do not know the appropriateness of OUMG for the tasks without any evidence. Second, only one poor model is selected as $G_2$. Are the results the same when the $k$ is changed more or less? What if $G_2$ takes $k+1$? Can OUMG show the positive HSRD value? I am curious about these results. Third, why is OUMG objective? How can we measure the objectiveness of OUMG? The authors need to show the evidence for their claim.

And the claim and idea are quite similar to the previous research work. [1-4] suggest trained evaluation methods for dialogue models. Especially, [1] suggests an adversarial learning method, and it is quite similar to the idea in this paper. This paper does not mention these related work.

More questions
- Even though human judgment has some bias, is there any positive relationship between OUMG and human judgment? Other evaluation methods show the relationship [1-4], so it would be better to make the same comparison.
- What is the value of alpha in equation (5) for the experiment? Are there any changes based on the alpha value?

Typos
- No ‘N’ in the equation (1)
- Would you please write the meaning of variables in equations (i.e., $X_1$ and  $X_2$ in the equation)


Reference
[1] Li, Jiwei, Will Monroe, Tianlin Shi, Sébastien Jean, Alan Ritter, and Dan Jurafsky. "Adversarial Learning for Neural Dialogue Generation." In EMNLP. 2017.

[2] Lowe, Ryan, Michael Noseworthy, Iulian Vlad Serban, Nicolas Angelard-Gontier, Yoshua Bengio, and Joelle Pineau. "Towards an Automatic Turing Test: Learning to Evaluate Dialogue Responses." In Proceedings of the 55th Annual Meeting of the Association for Computational Linguistics (Volume 1: Long Papers), pp. 1116-1126. 2017.

[3] Tao, Chongyang, Lili Mou, Dongyan Zhao, and Rui Yan. "Ruber: An unsupervised method for automatic evaluation of open-domain dialog systems." In Thirty-Second AAAI Conference on Artificial Intelligence. 2018.

[4] Bak, JinYeong, and Alice Oh. "Speaker Sensitive Response Evaluation Model." In Proceedings of the 58th Annual Meeting of the Association for Computational Linguistics, pp. 6376-6385. 2020.



**Summary Of The Paper:**

This paper presents a new evaluation metric for natural language generation models based on the discriminator that distinguishes between human-written texts and models generated texts, OUMG. To train OUMG, the authors bring two different generation models and treat them as negative samples. After the training OUMG, the authors suggest an evaluation method by computing the high score rate difference and upgrade rate among the two models. The authors also suggest guiding text generation by OUMG. Experiments show that trained OUMG can identify human-written texts against model-generated texts. And it can identify texts that a poor model generates.


**Summary Of The Review:**

Overall, I admit that OMUG is a sound method to identify some aspects of texts by humans and models. But I have some questions and concerns that I want to listen to the author’s responses.

---

> ### Author Response · Authors · 2021-11-21
> **Response to Reviewer Nb33**
>
> Thank you very much for your comments. we are sorry to have some writing omissions in the paper, which will be fixed in the next version. In this paper, we try to propose a universal and objective text generation metric.  The reason why we choose the Chinese poetry generation task is that the existing text generation metric is not ideal for the task. We want to prove the generality of our method by that. In order to further improve the persuasion, we will add some experiments on different tasks in the next version of the paper. Thanks again.

---

### Official Review · Reviewer_xKWL · 2021-11-03

**Correctness:** 2
**Technical Novelty And Significance:** 2
**Empirical Novelty And Significance:** 2
**Recommendation:** 3
**Confidence:** 3

**Main Review:**

- The idea of the proposed metric is somewhat intuitive and straightforward.
- The paper is structurally organized, although there are many missing/wrong explanations for notation (e.g. “X_1” and “X_2” from equation (1), “D” instead of “N” for the discriminator, etc.), which I suppose will be fixed in the next version of the paper.
- The paper can benefit from more justification and elaboration of when and why this metric, or more specifically the pairwise comparison, is useful. After reading the paper, it is still unclear to me when one would want to report numbers on the pairwise comparison (e.g. when you only care about two specific models? why?).
- The proposed method raises the question of reproducibility of the numbers reported on such metrics. In addition to providing more details on the experiment procedure (e.g. how did you sample sentences to compare? how many did you use?), the paper needs to prescribe a desired practice for using the proposed metrics: Should one fix one of the two models and swap out the other model to make consistent comparison among more than two models? Can one can reuse the discriminator trained on the two models to compute metrics for other models? How many samples do you need for training and testing? How robust are the metrics to sampling and the sample size?
- Also, defining a single number as an “objective” and “universal” metric for human-likeness calls for more qualitative analysis. It is unclear exactly what the metrics are capturing. The only explanation provided in the paper is “similar to humans.” More fine-grained analysis on various aspects of text (e.g. fluency, coherency, grammaticality, style, etc.) is desired and comparison with other evaluation metrics (not just conceptually, but quantitatively showing correlation) would be highly recommended.
- For the first part of the experiment, I am not quite convinced that the intentional degrading of the default model is the best way to demonstrate the effectiveness of the proposed metrics. It does show that the metrics can detect severely worse performance, but metrics are more valuable when they allow us to distinguish similar models in a more fine-grained way or to reason about specific aspects of quality. Also, for the second part of the experiment, I am not fully convinced with the claim that the dropped accuracy implies the improved performance of the guided model without knowing which sentences are sampled and what they are compared against (i.e. what has been used during training). Could it be a weird artifact due to some syntactic resemblance of sampled sentences at test time? More examples and qualitative analysis would help readers better interpret the results.
- Please consider including more information about resource requirement for training and discussing environmental impact. The proposed method seems to require quite expensive and time-consuming process of training and inference specific to the two models, or even a specific set of sampled sentences of interest. Even if one time evaluation cost of two models might be negligible, it is worth reflecting on the long-term impact.

**Summary Of The Paper:**

The paper proposes a new evaluation metric, OUMG, to measure human-likeness of machine-generated text without reference. The key idea of the method is to train a discriminator on both human-generated texts (as positive) and machine-generated texts (as negative) and use the discriminator to assign a score to a given sentence, which represents the similarity between the sentence and a set of sentences written by humans provided at training time.

The paper considers a pairwise comparison setting: any pair of models are compared by training such a discriminator on the outputs of those two specific models, and proposed metrics are used to measure the *relative* goodness of one model’s outputs with respect to the other model: (1) which model’s outputs have higher ratio of sentences with scores > 0.5 (i.e. HSRD) and (2) which model’s outputs have higher scores compared to the other model (i.e. UR).

The experiment considers the task of Chinese poetry generation and uses SongNet (Li et al., 2020) as a default model. The first half of the experiment intentionally makes the default model worse and demonstrates that OUMG metrics can show the default model performs better than the intentionally degraded model. The second half of the experiment uses the scores from the discriminator to adjust logits from the default model in order to improve the human-likeness of generation. The reduced accuracy of the discriminator after training on such outputs is interpreted as the evidence that the guided generation produces more human-like texts.

The main contribution of this paper is to frame text evaluation as pairwise comparison between two models and use relative metrics to measure the performance gap between two models. The authors claim that this approach addresses some of the limitations of previous metrics based on string matching, word embedding, and language models such as superficial matching, reliance on reference, narrow focus on fluency (Section 2).

**Summary Of The Review:**

The two main parts that are unclear to me are the need for *relative* metrics and what the metrics are actually capturing (supposedly similarity between human- and machine-generated texts). The experiment and analysis do not provide enough information to support the claim that the proposed metrics are objective/universal and have guiding ability.

---

> ### Author Response · Authors · 2021-11-21
> **Response to Reviewer xKWL**
>
> Thank you very much for your comments. we are sorry to have some writing omissions in the paper, which will be fixed in the next version. In this paper, we try to propose a universal and objective text generation metric.  The reason why we choose the Chinese poetry generation task is that the existing text generation metric is not ideal for the task. We want to prove the generality of our method by that. In order to further improve the persuasion, we will add some experiments on different tasks in the next version of the paper. Thanks again.

---

### Decision · Program_Chairs · 2022-01-20

**Decision:**

Reject

**Comment:**

The authors propose a reference-less metric for evaluating NLG systems by training a discriminator which distinguishes between human-generated and machine-generated text.

The main concerns raised by the reviewers were (i) lack of clarity in certain portions of the paper (ii) lack of demonstration of the "universal" applicability of the proposed metric (only evaluated for poetry generation) (iii) lack of clear guidelines on how to use the proposed metric in a reproducible manner (iv) lack of details about what exactly does the proposed metric capture and look for in the generated text.

The authors did not respond to the specific queries of the reviewer and agreed that more work is needed on their part.